# Method for simultaneous tracking of thousands of unlabeled cells within a transparent 3D matrix

Falk Nette[1,℮], Ana Cristina Guerra de Souza[1,℮], Tamás Laskay[2], Mareike Ohms[3], Daniel Dömer[2], Daniel Drömann[4], Daniel Hans Rapoport[5]*

1 Fraunhofer Research and Development Center for Marine and Cellular Biotechnology, Lübeck, Germany, 2 Department of Infectious Diseases and Microbiology, University of Lübeck, Lübeck, Germany, 3 Research Department Virus Immunology, Leibniz Institute for Experimental Virology, Hamburg, Germany, 4 Medical Clinic III Pneumology, University Medical Center Schleswig-Holstein, Lübeck, Germany, 5 Institute for Medical and Marine Biotechnology, University of Lübeck, Lübeck, Germany

℮ These authors contributed equally to this work.
* daniel.rapoport@uni-luebeck.de

**Data Availability Statement:** All relevant data are within the paper and its Supporting information files.

## Abstract

Three-dimensional tracking of cells is one of the most powerful methods to investigate multi-cellular phenomena, such as ontogenesis, tumor formation or wound healing. However, 3D tracking in a biological environment usually requires fluorescent labeling of the cells and elaborate equipment, such as automated light sheet or confocal microscopy. Here we present a simple method for 3D tracking large numbers of unlabeled cells in a collagen matrix. Using a small lensless imaging setup, consisting of an LED and a photo sensor only, we were able to simultaneously track ~3000 human neutrophil granulocytes in a collagen droplet within an unusually large field of view (>50 mm²) at a time resolution of 4 seconds and a spatial resolution of ~1.5 µm in xy- and ~30 µm in z-direction. The setup, which is small enough to fit into any conventional incubator, was used to investigate chemotaxis towards interleukin-8 (IL-8 or CXCL8) and N-formylmethionyl-leucyl-phenylalanine (fMLP). The influence of varying stiffness and pore size of the embedding collagen matrix could also be quantified. Furthermore, we demonstrate our setup to be capable of telling apart healthy neutrophils from those where a condition of inflammation was (I) induced by exposure to lipopolysaccharide (LPS) and (II) caused by a pre-existing asthma condition. Over the course of our experiments we have tracked more than 420.000 cells. The large cell numbers increase statistical relevance to not only quantify cellular behavior in research, but to make it suitable for future diagnostic applications, too.

## Introduction

Understanding multicellular phenomena and mechanisms such as ontogenesis, tumor formation, wound healing and cellular immune responses requires, among many other things, precise knowledge of the whereabouts and whenabouts of many thousand cells at once. Obtaining

**Funding:** Work carried out by FN and ACGS was funded by the European Regional Development Fund (https://ec.europa.eu/regional_policy/en/funding/erdf/) via the FIT-program of Schleswig-Holstein, Germany („KillAsthma", Grant-Number 123 17 024). The funders had no role in study design, data collection and analysis, decision to publish, or preparation of the manuscript.

**Competing interests:** The authors have declared that no competing interests exist.

this information can often be experimentally tedious [1], if not impossible. Therefore, most in-vitro experiments with tracking of individual cells have been performed in 2D-environments using some 10 to 100 cells only. Often, these 2D-environments are designed as microfluidic channels, which are convenient for optical observation and controlling the flow and distribution of nutrients and cell-stimuli. However, such experimental arrangements usually deviate heavily from the natural mechanical and chemical properties of the extracellular matrix which cells would experience in vivo. Therefore, in recent years, researchers have turned to investigate cells in three-dimensional environments for a wide variety of applications [2–5].

In this paper, we present a simple, yet powerful method for in-vitro 3D-tracking of many thousand cells at once with high spatial and temporal resolution and a large field (volume) of view. The method is demonstrated to work with 2000 to 5000 neutrophil granulocytes simultaneously, which are distributed three-dimensionally in a droplet of collagen, (4 mm to 5 mm diameter at the bottom and ∼1 mm height). The neutrophils were subjected to chemotactic stimuli (IL-8 and fMLP) as well as to different mechanical environments (stiffness and pore size of the collagen). Due to the high number of observed cells, the resulting 3D-tracks give a much more precise statistical account on the cellular response to these stimuli when compared previous methods, while at the same time being much easier to perform. We believe this increased statistical precision, together with the large field of view, the capacity to 3D-track cells and the overall simplicity of both, the optical and the assay setup to make for a novel and easy way to make three dimensional label-free cell tracking broadly accessible.

Technically, the method is a combination of in-line holographic microscopy [6] with digital reconstruction of the images on many different heights ("z-planes") and 3D-tracking the cells in this virtual z-stack. Dating back to Dennis Gabor's discovery of the holographic principle in 1948 [7], in-line holography is the simplest of all holographic setups: A beam of monochromatic and spatially at least partially coherent light is sent through the (translucent) sample. The portion of the light which passes through the cells becomes phase-shifted and diffracted with respect to the light which did not go through the cells (reference beam). Superposition of these different light paths yields diffraction patterns which can be recorded in the near-field of the sample (up to about 5 mm distance) using a conventional 2D photodetector (e.g. a CMOS-chip found in digital cameras). These diffraction images are the raw data of the experiments presented here. Numerical reconstruction can be used to compute a virtual z-stack of the three-dimensional arrangement of thousands of cells at once within this virtual z-stack, cells can be segmented and tracked thereafter using the open-source particle-tracking framework "Trackpy" [8].

In recent years, improvements in optical sensor technology as well as increasingly affordable computing power gave rise to digital holography as a feasible alternative to conventional bright-field time-lapse microscopy. Its potential for the three-dimensional tracking of live cells was recognized around the same time [9–12]. By 2007, Sheng et al. achieved the simultaneous tracking of 500 to 1000 dinoflagellates using an in-line configuration with a magnifying objective [13]. Later, Su et al. were able to accurately observe up to 1500 fast-moving sperm cells with a completely lens-free setup, leveraging dual-view illumination with two LEDs [14]. Living cells in culture were observed using a sophisticated digital holographic microscope in 2005 [15] and tracked in three dimensions [16]. Radically simplified lens-free in-line devices have been further proposed that can be placed inside an incubator to observe planar cell growth and/or movement over the course of hours or days [17–20]. Finally, these compact, simple set-ups have also been proposed to study cell growth in three-dimensional hydrogels: An optical setup very similar to ours was used to observe capillary growth in fibrin [21], though accurate positioning and tracking was not attempted; another group devised a 4-angle illumination and

reconstruction technique and were able to accurately position micro beads and organoids embedded in Matrigel® as a proof-of-concept [22]; a multi-color tomographic device was developed by Berdeu et al. [23] to track the growth and movement of endothelial cell clusters on a drop of Matrigel®. While this way of preparing a 3D cell culture is similar to our approach, the tomographic system, which needs physical rotation is too slow for tracking fast-moving cells, like neutrophils. These require frame rates in the order of a few seconds.

In this paper, we attempted to fuse the previous attempts into an simpler, yet even faster method to 3D-track cells. We demonstrate this setup with neutrophil granulocytes. These cells are fast movers and as such known to be the "first line of defense" in the cellular immune response of the human body. Neutrophils, which are by far the most abundant type of leukocytes in human blood, have developed several mechanisms to move through tissue and small pores to quickly reach their place of action [24, 25]. In particular, directed movement of neutrophils can be initiated through a variety of chemoattractive signal molecules (e.g., lipids, N-formylated peptides, factors of the complement-system). The directed migration along concentration gradients of chemoattractants is called chemotaxis. The distinct migratory behaviour, combined with the importance in the immune response make neutrophils an ideal model to establish new methods for fast 3D tracking of many cells.

Also, being able to track neutrophils in a simple assay opens up the possibility for new cell based diagnostics. It has been demonstrated previously that neutrophils of asthmatics tend to move slower along the concentration gradient of a chemoattractant [26]. However, these experiments were carried out in 2D microfluidic chambers and the tracking was performed with a few hundred cells only. Here, we demonstrate that our setup not only tracks at least ten times more cells in single experiment, but also that our simplified setup performs even better in discerning asthmatics from non-asthmatics.

Also, the method presented in this paper allows for easy variation of important cell migration parameters: Changing type, concentration and time sequence of chemokines; type, stiffness and pore size of the 3D-matrix are readily achieved; as well as variation of cell types and even making experiments with two or more different cell types.

# Materials and methods

## Assay

**Ethics statement.** Blood collection from healthy and asthmatic adult volunteers was performed with the understanding and written consent of each participant. The study was approved by the ethical committee of the University of Lübeck and the Fraunhofer EMB in Lübeck, Germany (ethics approval of the University of Lübeck no. 18–186 and 14–225).

**Isolation of human neutrophils.** Peripheral blood samples were collected by venipuncture from healthy donors using a lithium-heparin coated tube. Neutrophils were purified using discontinuous Percoll-gradient centrifugation as described in [27]. Neutrophil purity (>99.9%) was assessed by Giemsa staining. The viability of the cells (>99%) was determined by trypan blue exclusion. After isolation, neutrophils were resuspended in complete medium, namely RPMI-1640 medium supplemented with 10% heat-inactivated fetal calf serum (FCS), 4 mM L-glutamine, 10 mM HEPES, 100 U/ml penicillin and 100 μg/ml streptomycin (all from Sigma-Aldrich, Germany).

**Migration assay.** Of the above described cell solution, 50 μl with a cell concentration of 1 x $10^6$ cells/ml were mixed with 10 μl 10 × MEM, 14.4 μl $H_2O$, 3 μl 7.5% $NaHCO_3$ and 72.6 μl bovine collagen type I solution (3.1% PureCol, Advanced BioMatrix) to yield a total of 150 μl mastermix containing cells, 1.5 mg/ml collagen and medium.

A droplet (10 μl) of the cell containing mastermix-solution solution was placed into a ∅35 mm dish with a 180 μm thin polymer bottom (Ibidi μ-dish). This particular dish was chosen for the superior optical properties of its foil bottom which also allows to place the sample closer to the imaging sensor, resulting in a better imaging of the diffraction patterns. The collagen droplet was 4 mm to 5 mm in diameter and 1.2 mm to 1.5 mm high. It contained around 3000 cells, each of which was individually detected and tracked.

Chemotactic gradients were generated through diffusion of the chemoattractant from the surrounding medium into the collagen droplet. We investigated the shape, width and stability of the gradients using covalently rhodamine labeled dextran (MW 10 kDa, Sigma-Aldrich, R8881), the molecular weight of which is roughly similar to interleukin-8 (8.4 kDa), a method which has already been used to investigate diffusion in hydrogels [28, 29]. In particular, we filled a microfluidic channel (Ibidi μ-slide VI) with collagen-I-solution (2 mg/ml Purecol, Advanced BioMatrix) from one side and after 30 min of gelation time, added dextran-rhodamine solution (10 mM) from the other side. The diffusion front was observed using a confocal microscope (Zeiss LSM 510) at time intervals of 5 min. Approximately, the gradient zone extended from 600 μm to 800 μm within 20 min, which was the average time of our tracking experiments (cf. S1 Fig). We can therefore be certain that the chemotactic gradients are stable over the course of our tracking experiments.

In our first set of tracking experiments we tested the experimental setup with respect to regular chemotaxis. The chemoattractants used were interleukin-8 (IL-8, R&D Systems) at 1, 10, 100 and 1000 ng/ml and N-formyl-l-methionyl-l-leucyl-phenylalanine (fMLP, Sigma-Aldrich) at 1, 10 and 100 nM. The chemoattractants were dissolved in PBS and added to the above described complete RPMI-1640 medium to achieve the final concentrations (1 μl solution to 1 ml medium; controls were done with PBS only). To allow for proper gelation of the collagen, the closed dish was incubated at 37˚C and 5% $CO_2$ for 25 min. After gelation, the collagen droplet was overlayed with 1 ml of the medium, which either contained the chemoattractant or just PBS (control). Once the dish was filled with medium, time-lapse recording was started immediately.

Next, we varied the concentration of collagen to quantify the influence of pore size and stiffness of the matrix on migration behaviour. We used two different stock solutions for high and low collagen concentrations: 1.0, 1.5, 2.0 mg/ml from a lower concentration stock (Purecol, Advanced BioMatrix, 3.1 mg/ml) and 3.0, 4.0, 5.0, 6.0 mg/ml from a higher concentration stock (Fibricol, Advanced BioMatrix, 10 mg/ml). Gelation conditions were the same in all cases (see above). In these experiments, a constant IL-8 concentration of 100 ng/ml in the complete RPMI-1640 medium was used as a chemoattractant.

We also inverted the concentration gradient of the chemoattractant, thus creating a falling slope towards the outside of the droplet by adding the chemoattractant to the collagen but not to the medium (IL-8: 100 ng/ml or fMLP: 10 nM). After the gelation process, 1 ml of chemoattractant-free medium was added. Depletion of the chemoattractant occurred through diffusion along the concentration difference, i.e. out of the collagen matrix and into the medium, effectively establishing a reversed gradient.

To investigate chemokinesis, i.e. stimulation without a gradient, we added the chemoattractants in same concentrations to both the cell-containing collagen mix and the surrounding medium (IL-8, 100 ng/ml or fMLP, 10 nM).

It is well known that lipopolysaccharide (LPS), a strong pro-inflammatory substance from the outer membrane of bacteria, reduces the ability of neutrophils to migrate along an IL-8 gradient [30]. To gauge the ability of our setup to detect a condition of inflammation (and therefore its diagnostic potential), we pre-incubated neutrophils for 30 min with LPS (E. coli 0111:B4, Sigma-Aldrich; 1 ng/ml or 100 ng/ml at room temperature). Afterwards, the cells

were embedded in collagen as described above (without removing the LPS) and either tracked without further stimulation (control) or subjected to a chemoattractant (IL-8, outward gradient with 100 ng/ml).

All tracking experiments were carried out at 37˚C and in 5% $CO_2$ humidified atmosphere (incubator Binder CB 160). Holographic images were recorded every 4 s for a time period of 20 min. Blood samples from healthy donors were received from a anonymized pool of 15 persons; we started every set of tracking experiments with a control (no stimulation) to account for donor variations.

## In-line holography and cell tracking

Fig 1 depicts our optical setup along with a schematic of the migration assay. The optical arrangement is a very simple lensless holographic setup as described by Repetto et al. [31]. Partially coherent light is produced using a bright LED-source (peak wavelength 525 nm), shone through a collimating lens array (F/2.2, focal length 27 mm) and a 10 µm-pinhole (Thorlabs P10D). For being able to run experiments in parallel, we used two different setups with LEDs of slightly different brightness (Cree XP-E2 and Hebei 05W580EGC). The distance between

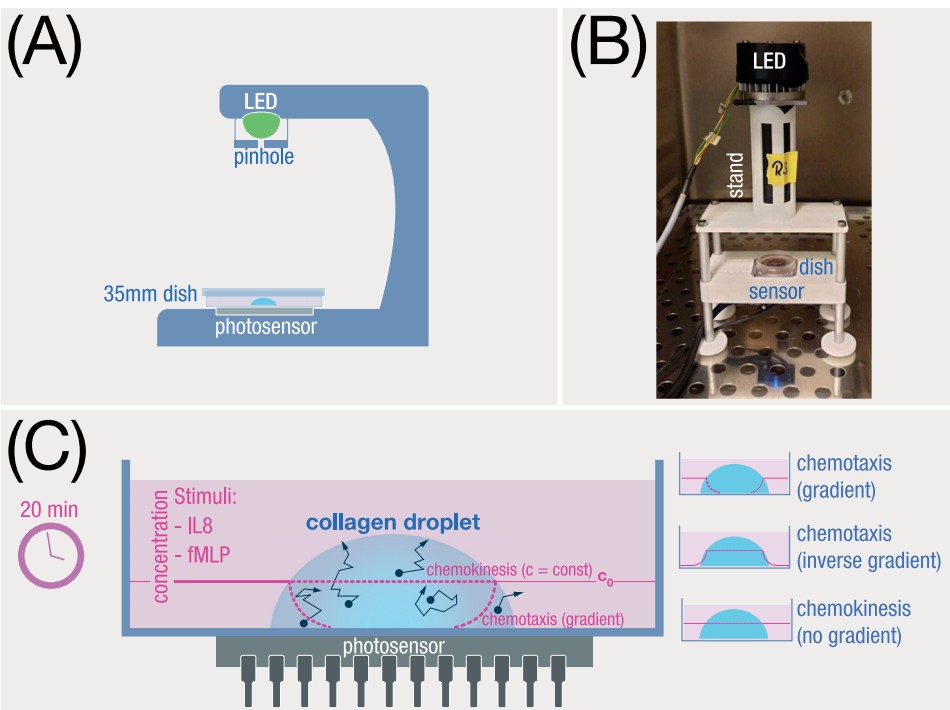

**Fig 1. (A)** Schematic view of the in-line holographic setup. A bright green LED produces spatially coherent light through a pinhole (∅10 µm in diameter). The light travels through phase objects (cells) at a different speed and direction than the undisturbed light, thereby producing diffraction patterns in the near vicinity. These patterns are recorded using a CMOS-sensor (8.6 x 6 mm, 42 MP) and later converted into 3D-images using a computer algorithm. **(B)** Photograph of a prototype of the holoscope in the incubator. The whole microscope is about 25 cm high, 12 cm wide and 6 cm deep; therefore, it easily fits into an incubator. **(C)** Submerged droplet assay: Neutrophils are embedded in a porous collagen droplet and subjected to different kind of chemokines (IL-8, fMLP). We investigated chemotaxis (i.e. cell movement in an outward gradient with respect to the collagen droplet), as well as chemotaxis in an inversed gradient (inwards with respect to the droplet). In another type of assay, we looked at the general, undirected increase of motility through chemokines but with no gradient between droplet and the surrounding medium (chemokinesis). In yet another set of experiments we varied the stiffness and the pore size of the collagen matrix using different collagen concentrations.

pinhole and sensor was 65 mm, which represents a good tradeoff between brightness and parallelism of the light. The exposure time was chosen to be 64 ms for the Cree LED and 75 ms for the Hebei LED. Short exposure times are preferable to reduce the impact of mechanical vibrations which otherwise might blur the image. Holograms were recorded with a debayered CMOS sensor (Raytrix GmbH Kiel, Germany) at a true resolution of 7716 × 5364 pixels with a pixel size of 1.12 x 1.12 μm$^2$, resulting in a very large field of view (8.6 x 6 mm). The time interval between subsequent image acquisitions was chosen such that even fast cell movements would be resolved sufficiently. Since neutrophils rarely move faster than 30 μm/min [32], a frequency of 15 captures per minute was chosen. At this rate, a 20 min time series generates about 12 GB of raw data (i.e. holograms), which in most cases could be reduced by roughly 50% through cropping images to a region of interest (bounding box of the collagen droplet).

**Hologram reconstruction, cell segmentation and tracking.** Fig 2A displays schematically how we extracted cell coordinates from the holograms. Numerical reconstruction was performed by applying a GPU-accelerated implementation of the angular spectrum method [33]. For each hologram, a virtual z-stack was created by reconstructing the phase images at $n$ equidistant steps $z_{start}$, $z_{start} + z\Delta$, ..., $z_{start} + (n-1)^* z\Delta$, where $z_{start}$ represents the distance

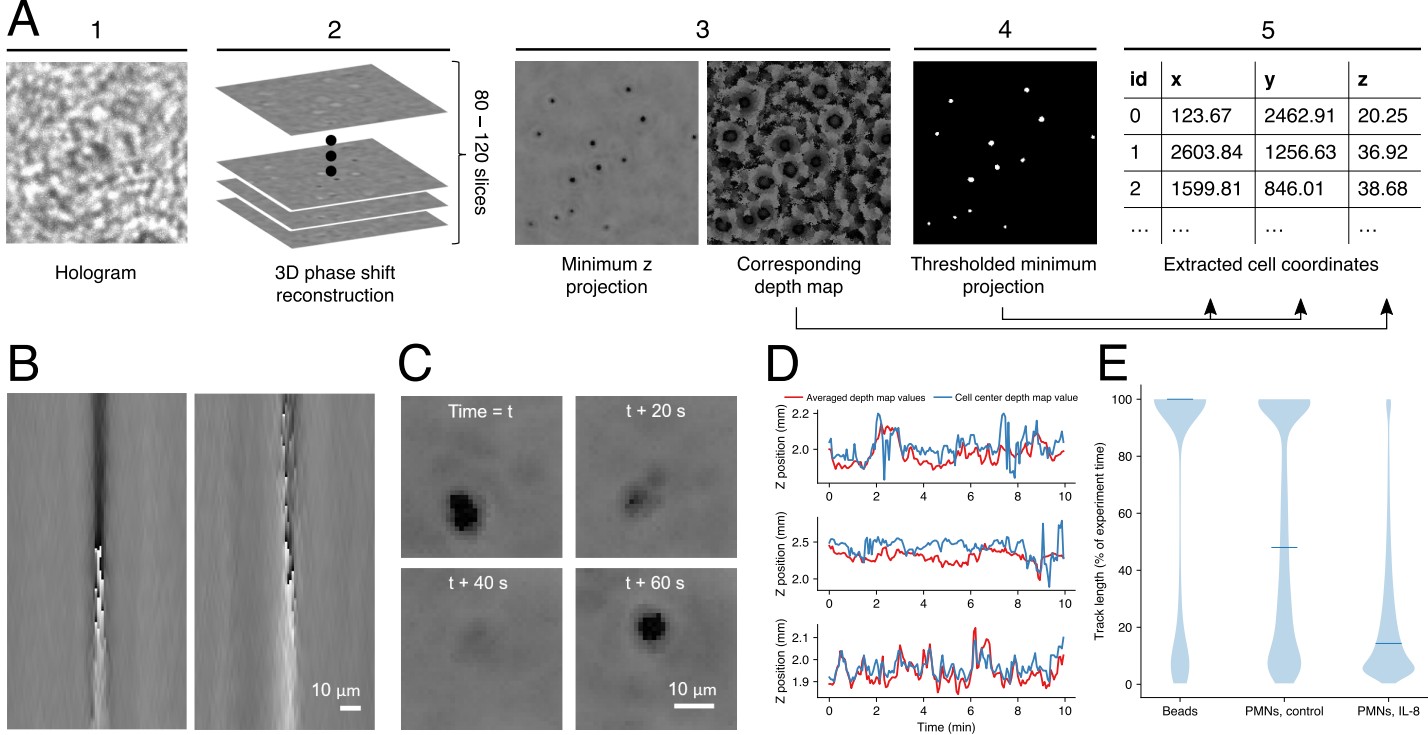

**Fig 2. Determination of cell positions and tracking accuracy.** (A) Cell coordinate extraction scheme (see text). (B) Phase-shift side(xz)-view of ∅15μm polystyrene beads in a collagen drop. The left image shows a bead that located near the rim of the drop and 1.7 μm above the sensor, the one in the right image was closer to the center of the drop (where the collagen layer is thicker) and 2.3 μm above the sensor. The field of view shown is 168 x 90 μm (height x width). (C) Phase shift minimum projections of one neutrophil granulocyte at four different time points, each 20 seconds apart, showcasing how the highly flexible morphology of this cell type can make it difficult to detect. (D) Z positions of three different cells, migrating towards an IL-8 gradient for 10 minutes. Two methods for determining individual z-coordinates are compared: averaging the z-coordinates of all cell pixels in the phase shift minimum projection (red graphs), and taking the z-coordinate of only the center pixel—rounded from the xy-centroid which is assumed as the cell position for tracking—as the z-position of the cell (blue graphs). (E) Distribution of track lengths for different tracking tasks. Blue lines represent respective mean track length. Left: Polystyrene beads are nearly static in the collagen and easily tracked throughout the whole time-lapse sequence (n = 3 independent experiments). Middle: Even without chemoattractant, some cells do move and also show a less consistent phase jump than beads, resulting in an average track length of half of the experiment time (n = 7). Right: Under the influence of 100 ng/ml IL-8 in the surrounding medium, most cells are moving fast; frequent collisions and changes in morphology lead to a lower average track length of about one sixth of the experiment time (n = 10).

of the reconstruction plane closest to the image sensor. This starting point was chosen to be immediately beneath the collagen droplet. Usually, it was located (depending on the experiment) between 1.4 mm and 1.8 mm above the sensor. We defined $z\Delta$ as the spatial sampling interval (slice thickness) of the z-stack. Choosing this value represents a compromise between practicability (larger $z\Delta$ means less computation time and memory/storage required) and exactness (smaller $z\Delta$ means a higher z resolution). Throughout all experiments, $z\Delta$ was set to 10 μm. With the neutrophil diameter averaging 6 to 8 μm [34], this does not theoretically fulfill the Shannon condition for accurate positional sampling in the z direction. However, the z resolution was lowered more severely by factors pertaining mainly to the system's point spread function as well as neutrophil deformability and composition (see Results section). Depending on the height of the collagen droplet, between 80 and 120 slices were reconstructed, resulting in reconstructed volumes of about 4x4x1 mm$^3$.

In the resulting volumetric data, cells appear excessively elongated along the z-axis and exhibit an hourglass-like shape, as illustrated in Fig 2B. Often, an abrupt jump from dark to bright pixels is visible where light passes the geometric focus of the cell, a phenomenon known as Gouy phase anomaly [35]. In [36] it is demonstrated how this characteristic phase jump can be used to determine the accurate positions of spherical particles along the z-axis. However, since cells—and neutrophils in particular—can assume a variety of non-spherical shapes, not all cells exhibit a clear phase jump. Still, we always found a minimum in the middle of the hourglass-like cell shapes (just not always immediately followed by a maximum). Therefore, we used the simplified approach of Sheng et al. [11], who took the minimum phase value along the z-axis to locate translucent polystyrene beads. Using the minimal phase value for determining the z-coordinate has proven to be reliable and computationally efficient also for cells. Mathematically, this approach can be expressed as

$$MinProj(x, y) = \min_{z_1,\ldots,z_n} Vol(x, y, z) \tag{1}$$

i.e. the reconstructed volume can be reduced to a minimum projection, which contains all darkest pixels along the z-axis for each location $(x, y)$. In practice, this was implemented such that already during reconstruction we looked for minimal values on the current z-plane and updated a minimum projection image iteratively. At the same time, when a pixel was updated with a new, lower value, we stored the z-distance of the current plane in a separate depth map. Thus, the final outcome of our reconstruction algorithm is not a time series of volumetric data, but rather a time series of minimum projections and depth maps. The reduction from ~100 to two images only results in a (lossy) compression rate of about 98%. Not only does this allow to use storage space more effectively but it also simplifies image processing and viewing.

Cells were detected by thresholding the minimum projections at each time point to separate foreground pixels (value $\leq$80, cells) from background pixels (value >80). Applying a fixed-value threshold to phase shift images works comparatively well, because the background (of mostly undisturbed light) occupies a fairly small range of gray values that is completely independent of changes in light intensity; these typically occur due to uneven illumination by the light source, differing light sources between devices, or different specimen with varying absorption characteristics. The centroid of each resulting connected foreground-component was computed in sub-pixel resolution and saved as a cells xy-position. Defining the z-coordinate of a given cell is not as straightforward as to consider the depth map entry at its cells xy-position, because the migrating neutrophils often assume irregular shapes. Therefore, we defined the z-position of a cell to be the average z-value of all pixels in its connected foreground-component. In Fig 2A, panel 3 shows the minimum projection and corresponding

depth information. Panel 4 shows the foreground-components (white pixels) resulting from thresholding the minimum projection. For each component, its corresponding area in the depth map was considered and averaged in order to obtain a scalar z value. The averaging approach reduces uncertainty of the z-measurement, making for smoother, more realistic cell tracks, as shown in Fig 2D, where both methods are compared to one another for three exemplary cell tracks.

As is usually the case with in-line holographic setups, the precision of locating the z-position turned out to be roughly one order of magnitude lower than the precision of locating cells in the lateral xy-positions (about 20 μm to 30 μm in z vs. 1 μm to 2 μm in xy, see Results). Mostly for that reason, we tracked the cells on the minimum projection time series only, thus ignoring large parts of the z-information. Still, the information about 3D-movement is contained within the data; for instance, we analyzed the mean migration in z-direction by averaging all z-positions over time (see Results).

Tracking (i.e. linking cell positions through time) was done using Trackpy 0.4.2 [8], a flexible, open-source Python implementation of the popular Crocker-Grier algorithm [37]. Basically, the algorithm calculates the probability for a cell to be the next-frame successor of a given cell by means of their mutual distance. We used a search radius of 10 pixels, equaling a maximum cell speed of 168 μm/min. The remaining Trackpy parameters were chosen as follows: `memory` was set to 4 frames; `adaptive stop` was set to 3; `adaptive step` was set to 0.95. The final output of the tracking algorithm is a mapping of the cell coordinates in each frame to a set of unique cell numbers.

All calculations were carried out on a computer equipped with an Intel Core i7–8700K CPU (at 3.7 GHZ), 64 GB RAM and a Nvidia GeForce RTX 2080 Ti graphics card. Reconstruction of the holograms was done on the GPU and takes about 6 hours for a typical experiment with 300 frames, cropped to a field of view of about $4000 \times 4000$ pixels, though depending on the final image size, 6 to 8 reconstructions could be run in parallel to speed up the process accordingly. Next, segmentation and extraction of cell coordinates took about 15 minutes on the CPU (parallel processes). Finally, the tracking algorithm ran on the CPU and took about 30 s to construct migration paths from the coordinates.

## Analysis of tracking results

The tracking results were used to quantify (I) the undirected speed and (II) the directional persistence of cell migration. More precisely, we calculated (I) the average undirected speed of all cells at a given point in time $\bar{s}(t)$, and (II) the average velocity of all cells towards the outer rim of the collagen droplet (i.e. along its radius) $\bar{v}_{outward}(t)$. The average undirected speed $\bar{s}(t)$ was calculated as

$$\bar{s}(t) = \frac{1}{\Delta t \times n} \sum_{i=1}^{n} \sqrt{\left(x_i(t+1) - x_i(t)\right)^2 + \left(y_i(t+1) - y_i(t)\right)^2} \qquad (2)$$

where $n$ is the number of cell tracks in frame $t$ and $(x_i(t), y_i(t))$ the coordinates of cell $i$ in frame $t$. Note that $n$ may slightly change over time, because the number of detected cells in the field of view is not constant (e.g. because of cells not being correctly detected; the average relative standard deviation of the number of tracks was found to be ∼4.2%, based on 93 experiments including all conditions presented here), and because short cell tracks <10 frames length are disregarded. The average outward velocity was calculated by measuring the change in distance

to the center of the collagen droplet per frame according to:

$$\bar{v}_{outward}(t) = \frac{1}{\Delta t \times n}\sum_{i=1}^{n}\sqrt{|x_{center} - x_i(t+1)|^2 + |y_{center} - y_i(t+1)|^2}$$

$$-\sqrt{|x_{center} - x_i(t)|^2 + |y_{center} - y_i(t)|^2}$$

(3)

The center $(x_{center}, y_{center})$ of the collagen droplet was determined by the center of the droplet-circumscribing rectangle. Both $\bar{s}(t)$ and $\bar{v}_{outward}(t)$ where plotted as a function of time to assess differences between differently stimulated neutrophils.

## Results

### Accuracy of the holographic setup and reliability of cell tracking

In a first set of experiments, we determined the accuracy of our holographic setup and the reliability of cell detection and tracking. The z-resolution of the system was measured using fluorescent microbeads (BD Biosciences "Calibrite" PMMA, ∅6μm) that were deposited below and above a glass coverslip. We have measured this height distance using a conventional laser scanning microscope and compared it to the results of our holographic setup (refer to S3 Fig). The z-positions in the holographic setup have a slightly higher standard deviation than the those of the reference, but the overall accuracy of our setup (with microbeads) is within ∼6% of the reference microscope (LSM: 120.9 μm average z-distance between upper and lower beads; holographic setup: 127.9 μm). These values give a good estimate for the precision of our holographic setup along the z-axis. However, in cell migration assays, the precision is further reduced because (I) unlike microbeads, cells are not perfectly spherical, homogeneous objects; (II) the surrounding collagen-matrix is not optically homogeneous, because of pores and density fluctuations which produce background diffraction patterns; (III) mechanical vibrations within an incubator (due to ventilator motors), which might blur the image. To assess the extent to which these factors reduce the precision of localizing cells, we have performed another experiment with larger beads (Polysciences GmbH "Polybead", polystyrene, ∅15μm), which were embedded in collagen and measured within an incubator. We had to use larger beads because the small beads of the previous experiment were found to be very hard to detect when embedded in the matrix; we take this as a hint that the method might not be applicable to very small cells in a non-homogeneous matrix like bacteria in collagen. As shown in Fig 2B, the positions of the larger, embedded beads are indeed less well defined; we estimate the precision that can be achieved from these reconstructions to be ∼30 μm in z-position and ∼1.5 μm in x and y-position. We take these values to be the real-world error for determining cell positions in our experimental setup.

Throughout all experiments, we found a residual in the undirected migration speeds $\bar{s}(t)$ of about 5 μm/min—even when imaging (stationary) microbeads. This evidences that the apparent movement originated from small fluctuations within the diffraction patterns. These in turn could have been caused by minute movements of cells/beads within the collagen matrix but also of the collagen itself due to mechanical and thermal noise. An additional smaller source of error is introduced through discretization of the xy-position/sampling errors of the imaging sensor. Applying a rolling average (i.e. a low-pass filter) to the displacement (frame to frame vector-distance in the xy-plane) caused $\bar{s}(t)$ to drop close to zero, proving that the observed residual speed is caused by small random fluctuations in the xy-coordinate measurements and can be regarded as noise. However, we chose to present our data in its original, unfiltered form.

Tracking reliability does not only depend on the accuracy of cell coordinate measurement, but to an even greater extent on the ability to find cells and to correctly connect them through time. We found the following detection and tracking errors to be most relevant [38]: (I) False positive detections, where artifacts are erroneously detected as cells; (II) False negative detections, where the threshold detection fails to detect an existing cell; (III) Collisions, where cell paths cross or come into close proximity such that they cannot be sufficiently told apart.

False positive detections were the least common problem. Mostly, they were caused by large objects (like air bubbles, filaments or cell clumps) and could be excluded manually before cell detection. Sometimes, these objects can move, but most often at speeds much larger than migrating cells, leading to them being excluded from tracking due to the algorithm's search radius cutoff. In addition, we introduced a minimum track length of 10 frames to filter out false positive artifacts. Even though they constitute the largest number fraction (50–60% of all tracks), these very short paths are the least reliable tracks and can therefore considerably obscure the real migration dynamics. False negative detections were partially addressed through the `memory` parameter of the tracking algorithm, which determines the number of frames a found cell is remembered for, in case it fails to be detected in subsequent frames. We have chosen the parameter such that it continues to search for successors if the cell becomes temporarily "invisible" to the cell detection algorithm for 4 consecutive frames before terminating a track. Finally, collisions are the most difficult to resolve type of tracking error. However, in this work, we only looked at the momentary values $\bar{s}(t)$ and $\bar{v}_{outward}(t)$ which are almost unaffected by errors through confusion of cells (as opposed to per-track-measurements, e.g. the mean speed of individual cells). Of course, other types of motion characteristics, which span longer time ranges (e.g. cell divisions and genealogic trees), would require an extra treatment of collision errors.

Tracking reliability can be assessed quantitatively by analyzing the distribution of track lengths. Fig 2E compares this distribution for (I) the above mentioned larger polystyrene beads (∅15μm); (II) unstimulated neutrophils and (III) neutrophils stimulated with 100 ng/ml IL-8. In all cases, we find a bimodal distribution, with maxima at both ends, i.e. the highest and the lowest possible track length. However, the ratio between these maxima shifts with object speed from mainly maximally long tracks for slower objects (beads) to mainly maximally short paths for faster objects (stimulated cells). This result directly reflects that tracking becomes more difficult the more collisions occur and the more object locations change between two consecutive frames. In addition, detection itself became more error-prone with faster moving neutrophils due to intermittently weak signals in the phase images, an example of which is shown in Fig 2C. This happens most likely because neutrophils assume elongated and irregular shapes when they move towards a chemoattractant [39].

## Chemotactic response to IL-8 and fMLP

Both chemoattractants (IL-8 and fMLP) were found to reliably induce directed cell migration towards rising concentrations. The results of an exemplary IL-8 experiment are shown in Fig 3. Neutrophils from the same donor were tracked for 20 min, both without stimulation (control) and with 100 ng/ml IL-8 mixed with the surrounding medium (outward gradient). When comparing cell tracks of the control (Fig 3A, left) with the stimulated cells (Fig 3A, right), an increased migration activity is obvious. The displacement plot in Fig 3B shows that the stimulated cells travel further from their starting point as well. The reconstructed holograms of the IL-8 containing experiment are shown in S1 Movie. The average cell speed $\bar{s}(t)$ was calculated from the paths in Fig 3A and is shown in Fig 3C (left). It rises to almost 25 μm/min after addition of the chemoattractant, while barely exceeding the noise floor of

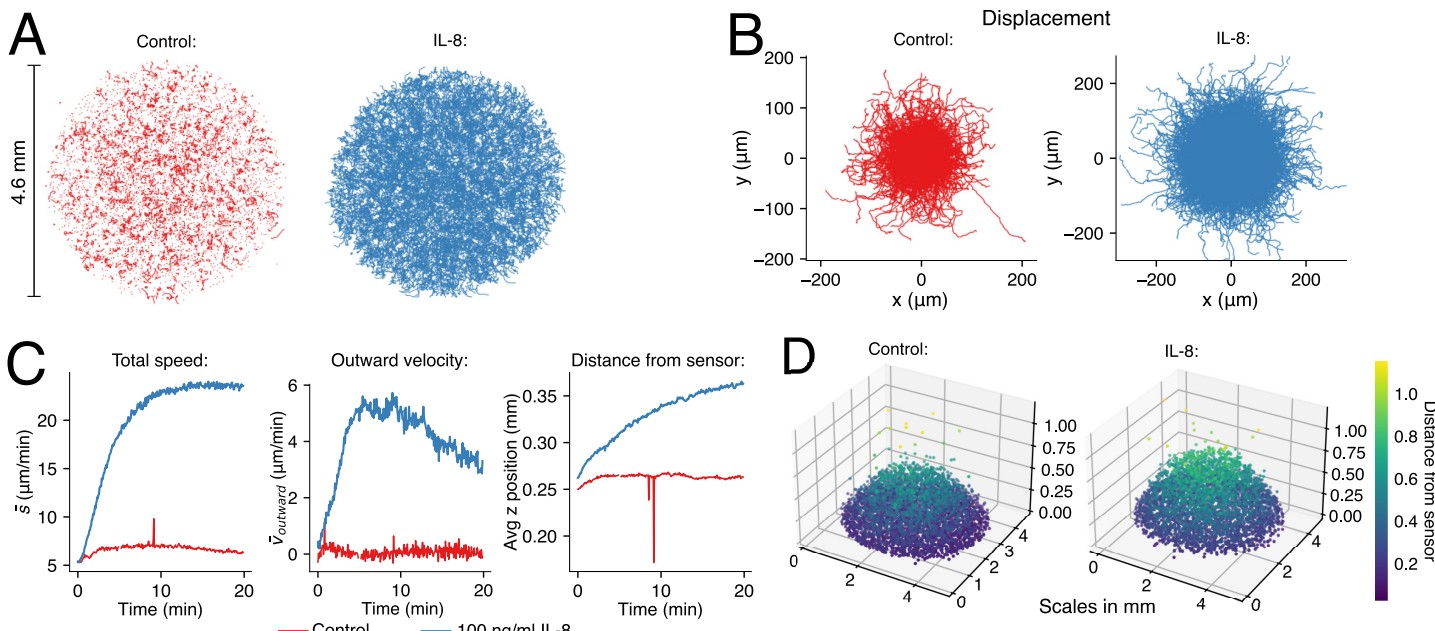

**Fig 3. Two example experiments.** Neutrophil migration control experiment (red) and with 100 ng/ml in the surrounding medium (blue), about 4000 cells each from the same donor. **(A)** 2D trajectories (xy-view), recorded over 20 minutes. **(B)** Rose plot of the same trajectories, each originating at (0,0) **(C)** Comparison of average speed, average outward velocity and average z-position. The spikes around the 10-minute mark of the control experiment are an artifact as a result of vibrations. **(D)** Individual cell positions after 20 min without and with 100 ng/ml IL-8 added to the surrounding medium.

∼5 μm/min in the control experiment. Fig 3B (right) displays the corresponding outward velocity $\bar{v}_{outward}(t)$ over time. As expected, the control reveals no preferred migration direction. This is reflected in the red $\bar{v}_{outward}(t)$-curve in Fig 3C (right), which fluctuates around zero. By contrast, the stimulated cells quickly start to move outwards, reaching a maximum of the outward component of the velocity of ∼5 μm/min after about five minutes. Though slowly declining after ten minutes, the value remains positive throughout the experiment, which is in line with our observation that the gradient remains stable over the course of the experiment (cf. S1 Fig). From these data, the chemotactic index can be calculated directly as the ratio between directed and undirected movement (25 μm/min peak for $\bar{s}(t)$, 5 μm/min of which are actually migration towards the rising gradient, resulting in a chemotactic index of 0.2). This agrees well with the chemotactic index values found in previous reports, though these tend to span a wide range between 0.1 and 0.9 [40–42].

Changing the concentration of the chemoattractant led to different migration speeds (see Fig 4A for IL-8 and Fig 4B for fMLP). Increasing the IL-8 concentration from 1 to 10 to 100 ng/ml caused the maximum migration speed $\bar{s}(t)$ to rise from 9 to 11.5 to 23 μm/min. When neutrophils were subjected to an even higher concentration of IL-8 (1000 ng/ml), the maximum speed decreased to 19 μm/min. A similar behavior was observed for fMLP. Here, increasing the concentration from 1 nM to 10 nM yielded maximum migration speeds of 16 and 22 μm/min, respectively. Again, further increase of the concentration (100 nM) lowered the maximum speed to 17 μm/min. We attribute the lowered chemokinesis to saturation and internalization of the respective chemokine-receptors (IL-8RA & IL8-RB; FPR1) [43–45]. Comparing reaction times, i.e. the time from starting an experiment until the maximum migration speed is reached, we found no clear indication that larger concentration differences (which correspond to higher diffusion speeds) lead to a faster cell response; processing time of

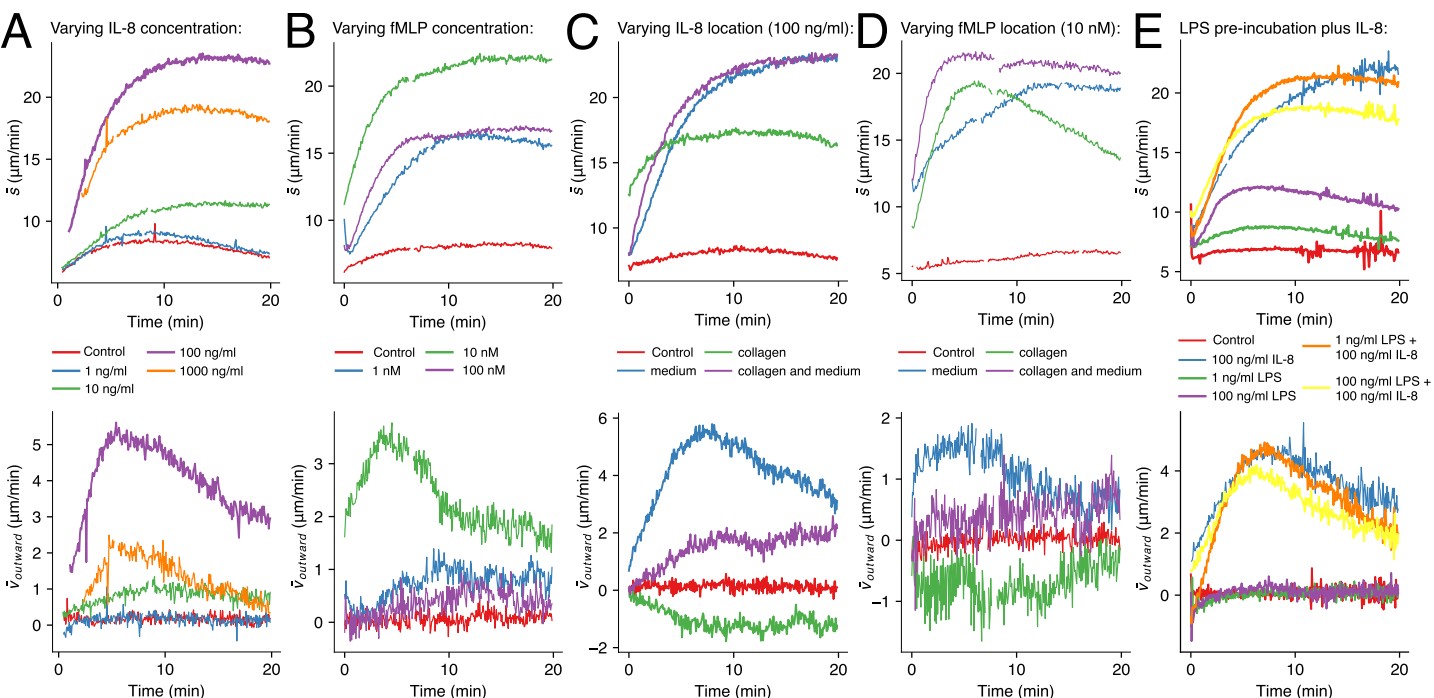

**Fig 4. Results of chemotaxis experiments.** **(A)** Average undirected speed (top) and average outward velocity (bottom) of neutrophils subjected to concentrations of 0, 1, 10, 100, and 1000 ng/ml IL-8 in the surrounding medium. **(B)** Same as (A) but for 0, 1, 10, and 100 nM fMLP. **(C)** Average undirected speed and average outward velocity of neutrophils subjected to a concentration of 100 ng/ml IL-8 in the surrounding medium (blue), in the hydrogel (green), in both (purple), without IL-8 (red). **(D)** Similar to (C) but with 10 nM fMLP. **(E)** Migration behavior of neutrophils is altered by incubation with LPS. Comparison of six conditions: low dose (1 ng/ml) and high dose (100 ng/ml) LPS, each with and without 100 ng/ml IL-8, control. Each graph in every figure displays the weighted average of three independent experiments.

the signal and turning it into motion seems to be largely independent of signal strength. The concentration dependence of the outward migration component $\bar{v}_{outward}(t)$ is displayed in Fig 4A (bottom) and Fig 4B (bottom). Similar to $\bar{s}(t)$ it displays a maximum at 100 ng/ml IL-8 and 10 nM fMLP, respectively. In case of fMLP, directed migration was found to be somewhat less pronounced than for IL-8, especially for concentrations higher or lower than 10 nM.

Reversing the direction of the gradient, i.e. depleting the chemoattractant towards the outside of the collagen droplet resulted in an overall reversed migration of the cells from the droplet rim towards its center. This proves that our assay really measures the effect of the chemoattractant. The results of these reversed migration experiments are shown in Fig 4D and 4E (green curves). As with the "normal" outward gradient, albeit somewhat less pronounced, the presence of chemoattractant led to an increased undirected migration speed $\bar{s}(t)$ (IL-8, outward vs. inward: 23 vs. 17 μm/min; fMLP, outward and inward: 19 μm/min, see Fig 4D (top) and Fig 4E (top)). The bottom graphs in Fig 4D and 4E show that the migration direction reverses with the gradient direction, indicated by the negative values for $\bar{v}_{outward}(t)$. The effect can be observed with both chemoattractants, IL-8 and fMLP, but again less distinctively with fMLP. As a final control, we have also stimulated the cells without any (macroscopic) gradient by adding the chemoattractant to both the collagen and the surrounding medium. As anticipated, this led to an increased undirected migration speed $\bar{s}(t)$, the magnitude of which was similar to the gradient experiments (IL-8: 23 μm/min; fMLP: 21 μm/min, see purple curves in Fig 4C and 4D). Unexpectedly, we also found a slight preference of the cells to move outside, away from the center of the collagen droplet. We hypothesize this small effect to be caused by

## Simulated inflammation: Incubation with LPS prior to chemotaxis with IL-8

Fig 4E (top, yellow curve) shows that pre-incubation of neutrophils with a high concentration of LPS (100 ng/ml) reduced the directed outward migration $\bar{v}_{outward}(t)$ by about 20% with respect to normal IL-8 stimulation (blue curve). The low LPS concentration (1 ng/ml, orange curve) had a much smaller effect on $\bar{v}_{outward}(t)$ and resulted in roughly the same curve as only IL-8 (blue curve). Results for the undirected migration speed $\bar{s}(t)$ were similar (Fig 4E top): Pre-incubation with a high-concentrated LPS leads to a reduced migration speed (19 µm/min vs. 22.4 µm/min max. speed, with and without pre-incubation). Again, small LPS concentrations (1 ng/ml) had no clear effect on migration towards the IL-8 gradient.

While LPS at the low concentrations (1 ng/ml) did not affect the subsequent response towards IL-8, it still had an effect on the undirected migration speed, which was ∼2 µm/min higher than the control (cf. Fig 4E top). This finding is in agreement with recent reports on the effect of super low dose LPS on neutrophil migration [46]. The high LPS concentration had an even stronger impact on the undirected migration speed, with $\bar{s}(t)$ being ∼5 µm/min higher than the control.

## Changing matrix stiffness and pore size through collagen concentration

Collagen was prepared at seven different concentrations (1, 1.5, 2, 3, 4, 5 and 6 mg/ml) and the cells were either stimulated with a IL-8 outward gradient at 100 ng/ml or not stimulated (control). Fig 5A (left) shows that the undirected migration speed $\bar{s}(t)$ drops by a factor of ∼2 when raising the collagen concentration by a factor of 6 (23 µm/min at 1 mg/ml vs. 12 µm/min at 6 mg/ml). The outward component $\bar{v}_{outward}(t)$ drops by a larger factor of 5 (5 to 1 µm/min for 1.5 and 6 mg/ml collagen, respectively; Fig 5A, right). The overall slowdown of the cell migration is most likely due to decreased pore sizes in higher concentration collagen and was found to be in line with reported values for unstimulated cells [47].

Fig 5B displays a two-dimensional histogram of the distributions of momentary cell speeds over the course of 20 min, for low and high collagen concentrations (1.5 mg/ml, left and 6 mg/ml, right). These visualizations reveal considerably more information than the average values $\bar{s}(t)$, shown in Fig 5A. In particular, it becomes clear that cells, if not hindered, assume a much greater variety of different speeds than in a motion-restricting environment (variance of speed of ∼20 µm/min vs. ∼10 µm/min at 1.5 and 6 mg/ml collagen, respectively).

Varying the collagen concentration with unstimulated cells, resulted in a different behavior. Here, we found an almost unaltered noise floor of ($\bar{s}(t)$ at ∼5 µm/min (cf. Fig 5C, left). The increased apparent mobility of the cells in the lowest collagen concentration is most likely due to the decreased stiffness of collagen, which at 1 mg/ml was at the brink of becoming a viscous liquid. The corresponding distributions of momentary cell speeds (cf. Fig 5D) are much narrower and less dependent on the collagen concentration compared to the experiment with stimulated cells. This proves that the broad distribution of migration speeds is a consequence of the stimulation, rather than of not being motionally restricted by small pores.

In addition to migration speed, collagen concentration profoundly influences motility patterns of leukocytes travelling towards a chemotactic gradient [47]. We have measured the directionality of cell movement by looking at the distribution of migration angles. For each three successive cell positions along a migration-path, the enclosed angle was computed according to $arccos(\hat{u}_{1,2} \cdot \hat{u}_{2,3})$, where $\hat{u}_{1,2}$ and $\hat{u}_{2,3}$ denote the unit vectors between two

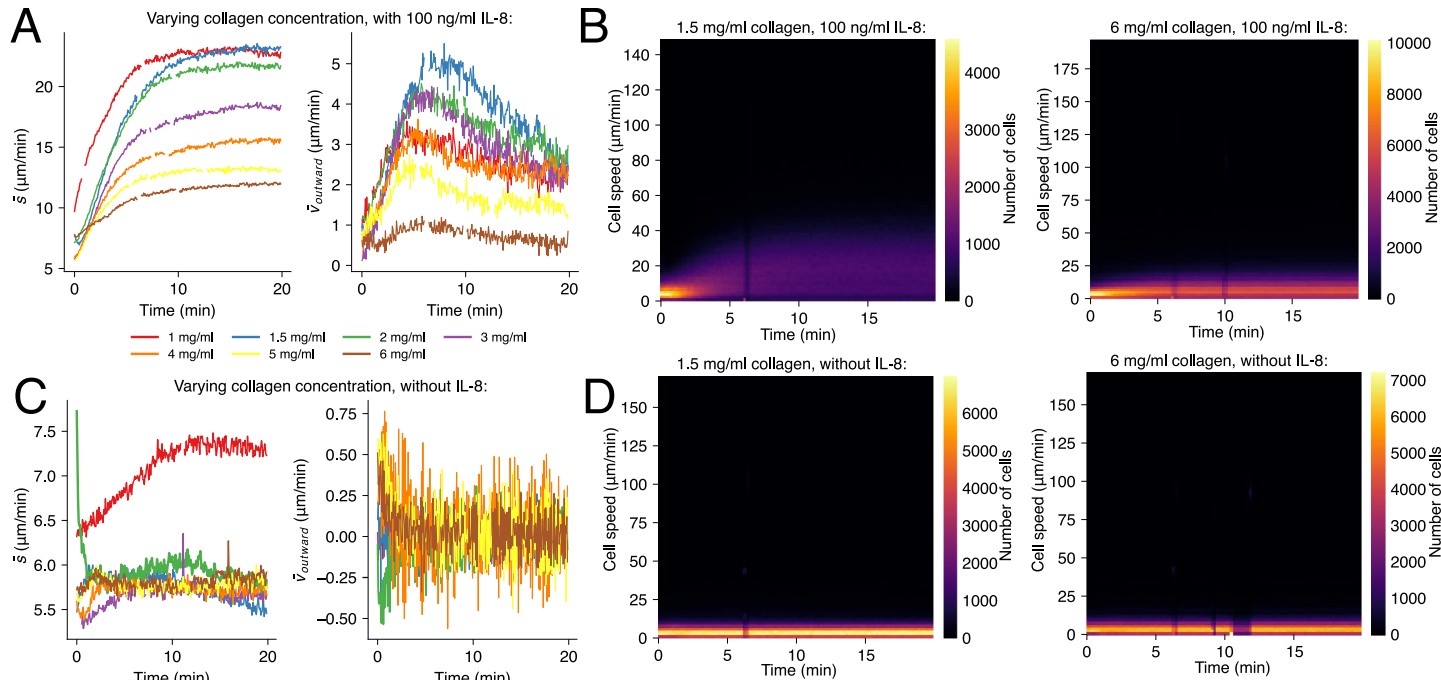

**Fig 5. Collagen concentration measurably influences spontaneous migration speed and chemotaxis velocity.** (A) Average cell speed and velocity for all collagen concentrations with 100 ng/ml IL-8 added to the surrounding medium. (B) Two-dimensional histograms of measured neutrophil speeds in low (left) versus high concentration collagen, with 100 ng/ml IL-8. A lighter color represents a higher number of cells moving with a specific speed at a specific time. (C) Speed and velocity for all collagen concentrations without chemoattractant. (D) Two-dimensional histograms of measured neutrophil speeds in low (left) versus high concentration collagen, without chemoattractant. Each graph in every figure displays the weighted average of three independent experiments.

respective positions. For each collagen concentration listed above, the relative frequencies of migration angles are shown in Fig 6. They can be understood as the probability of a cell to either keep its direction ($\sim$0 rad), to deviate from it (0 to $\pi/2$ rad) or to turn around ($\pi/2$ to $\pi$ rad). In case of chemotactic stimulation (blue histograms in Fig 6), the angles around 0 rad dominate, indicating that the cells movements have a significant amount of determination. However, as the collagen concentrations increase (and pore sizes decrease), the migration paths become more irregular, because cells enter obstacles more often. In particular, the number of turning events ($\sim\pi$ rad) rises abruptly at collagen concentrations >4 mg/ml (Fig 6F and 6G). This finding is in line with results presented by Francois et al. [47], which similarly indicate a threshold in collagen concentration above which both the amount of randomness in the migration patterns as well as the number of turning events increased markedly. This sudden—rather than gradual—increase is possibly due to transitioning from the dilute to the semi-dilute domain of collagen solutions and the accompanying structural changes around $c_{coll}\sim$5mg/ml [48]. By contrast, the angle distribution of unstimulated neutrophils displays a maximum at $\sim\pi$ rad (turning events, cf. red histograms in Fig 6). In the absence of a chemoattractant, the majority of steps seems to be back and forth, regardless of collagen concentration. Still, as with stimulated cells, the relative amount of turning events rises with increasing collagen concentration.

## Measuring chemotactic response of asthmatic neutrophils

Neutrophils taken from asthma patients have been previously demonstrated by Sackmann et al. to move at reduced migration speeds towards an fMLP concentration gradient [26]. In

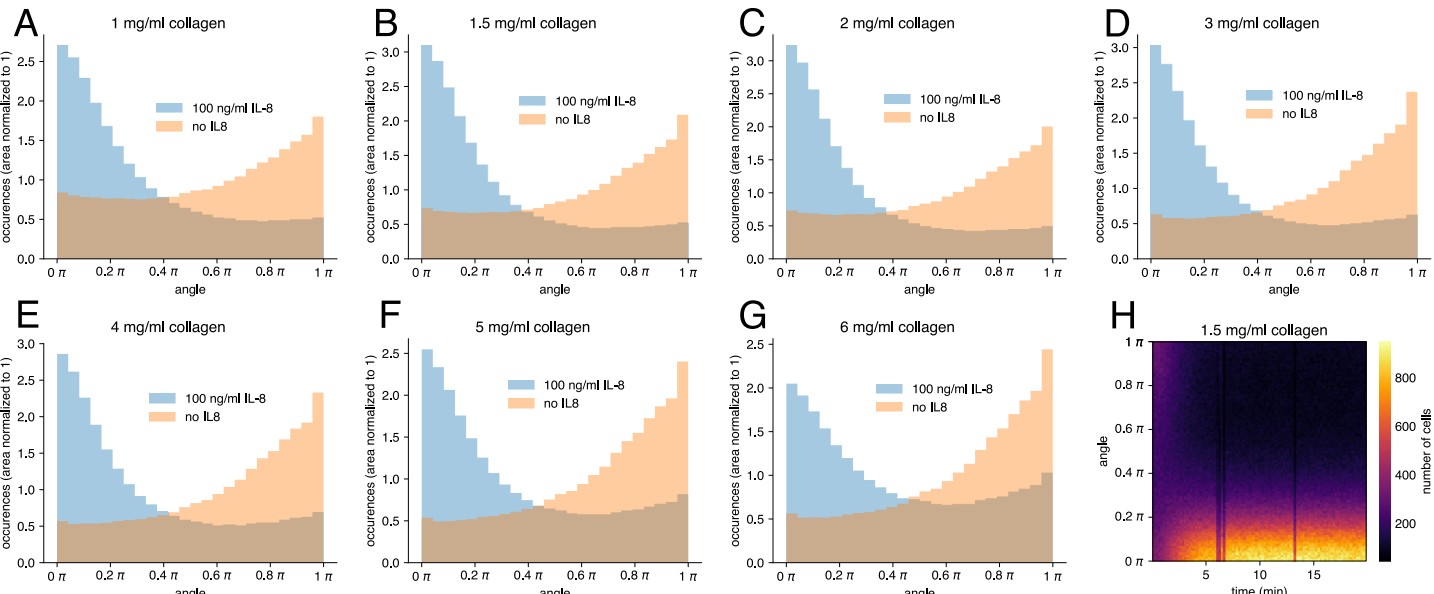

**Fig 6. Distribution of migration angles throughout the experiments shown in Fig 5A.** (A)-(G): Cumulative histograms showing the distribution of migration angles of cells migrating in collagen environments of different concentrations along an IL-8 gradient (blue) and without stimulation (red), experiment time 20 min (**H**): Two-dimensional histogram of migration angles against experiment time for a collagen concentration of 1.5 mg/ml within a 100 ng/ml IL-8 gradient. The graph shows that directed movement of the cells sets in over a time period of ~5 min. Each graph in every figure displays the weighted average of three independent experiments.

their work, they used a clever microfluidic system which allowed for sorting neutrophils from whole blood via adherence to a P-selectine coated surface; and subsequently observing the migratory behaviour of these neutrophils in an fMLP-gradient. Here, we demonstrate the suitability of our setup to perform the same task, i.e. to tell apart neutrophils from asthmatic and non-asthmatic individuals via quantifying a chemotactic assay. In total, we did twelve experiments to compare two conditions (reaction of asthmatic neutrophils to IL-8 and fMLP). We collected blood from five asthmatic donors, one sample was used for both, IL-8 and fMLP. The results were compared to six control experiments, for which blood was collected from six healthy (non-asthmatic) donors, resulting in a total four groups with N = 3 donors each (IL-8, fMLP and two control groups).

Fig 7 displays the results of these experiments using IL-8 and fMLP, respectively. In case of IL-8, the undirected cell speed was observed to be higher for neutrophils from asthmatics than from non-asthmatics. Despite the elevated overall migration speed, the velocity of directed movement along the gradient was lowered with respect to non-asthmatics (mean values: 2.87 vs. 3.98 μm/min). In case of fMLP, both undirected and directed movements were found to be slowed compared to non-asthmatics, though the reduction in undirected speed is less pronounced than under IL-8 influence. The reduction in chemotaxis velocity of about 32% (mean values: 1.58 vs. 2.34 μm/min) agrees well with the 27% reported by Sackmann et al.

## Discussion

In this paper, we present a simple and inexpensive method which allows for simultaneous 3D tracking of thousands of unlabeled cells in a large volume at high time resolution in vitro. We have shown it to be applicable to quantify the influence of various important factors affecting cell migration, such as cell type, matrix stiffness and pore size, type and concentration of chemokines, as well as influence of artificial and pathological inflammation conditions. These

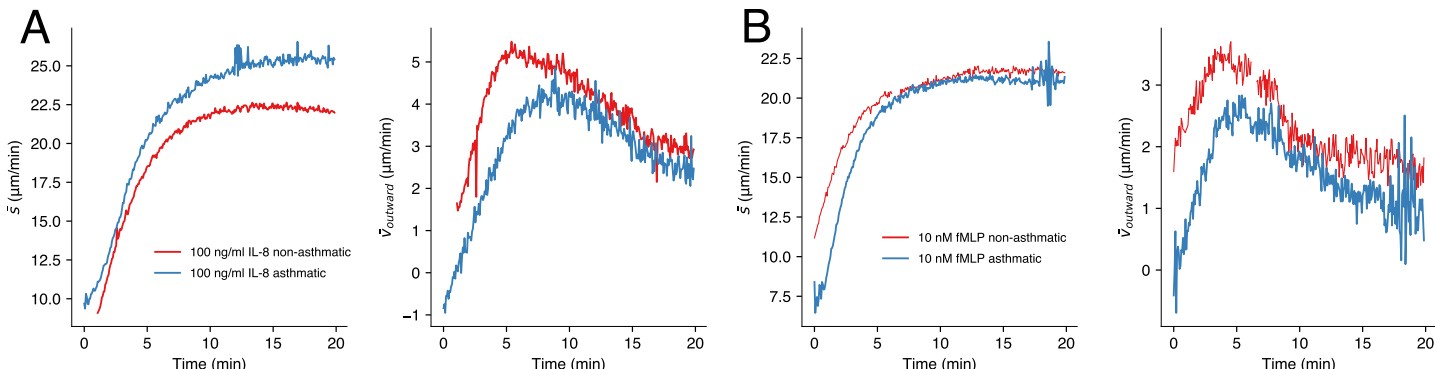

**Fig 7. Reduced migration speed of neutrophils from asthma patients towards chemotactic gradients of (A) IL-8 and (B) fMLP.** While the undirected cell speed can be elevated in asthma patients (IL-8, left graph), the directed outward movements is slower at any time and for both chemoattractants, IL-8 and fMLP. Each graph in every figure displays the weighted average of three independent experiments.

kinds of investigations are not only useful for understanding the migration of immune cells, but for any multicellular process in general, like the metastasis of tumor cells or wound healing. All of these experiments can be done in a well controllable 3D environment, which can be tailored to the scientific question at hand.

We also want to point out that to the best of our knowledge, despite its simple construction, no other experimental method achieves similarly large fields of view (actually, large volumes of view), combined with a very high time resolution. For instance, in a recent work François et al. [47] tracked more than 20.000 neutrophils in collagen with frame rate of 30 seconds. While these numbers are impressive and at the forefront of what is possible using conventional and laser scanning microscopy, we have, by comparison, investigated ~420.000 cells with a time resolution of 4 seconds. This represents an improvement of roughly one order of magnitude in time resolution and statistical relevance.

Our method, however, also has some limitations, the most obvious being that in-line holography works at low to medium cell densities only (order of magnitude: $10^3$-$10^6$ cells/ml). This is a direct consequence of not using a separate reference beam for holography, but instead relying on the condition that a large enough fraction of the light does not hit a cell, thus serving as a reference beam. For higher cell densities, as found in most tissues and tumors, not only the loss of reference beam becomes problematic, but also the increasing opacity of the sample. Therefore, the best suited assays for our holographic setup should use a transparent 3D matrix with some $10^4$ cells in a ~5 x 5 x 5 mm$^3$ volume. Within these bounds, the method provides a uniquely simple, yet powerful way to track cells in three dimensions.

Another limitation of the method pertains to the fact that the image sensor on which the cell dish is placed may run hot over time. It has been reported that this heating can lead to problems [23], both with the stability of the assay and the viability of the cells. We have measured the temperature of the camera PCB using an infrared camera and found one component (not the sensor itself) to heat up to 49˚C. This may pose a problem, in particular if the sample/ culture dish has a small volume, which may readily heat. It may also be problematic in longer time-lapse experiments (hours to days). However, in our experiments, which used a volume of ~1 ml of medium, we found neither a significant heating of the medium above 37˚C (cf. S2 Fig), nor any signs of a detrimental effect of temperature within the 20 min of time-lapse recording. In case of longer observations and/or smaller volumes, it may still be necessary to either actively cool the electronics similar to Berdeu et al. [23], or alternatively to shut the camera off when not needed.

The method is also an improvement over two dimensional setups (using e.g. microfluidic channels) in which cells cannot move in three dimensions at all. To a large degree, these 2D-setups are state of the art for investigating cell migration [49]. This seems to be mostly due to the fact that these devices can be constructed and controlled much more easily than 3D assays. Techniques for building custom microfluidic devices have become available to biologists, while at the same time conventional microscopy has been extended to easily allow for time-lapse recordings; tracking algorithms have been improved as well. However, although these experimental setups have led to a wave of migration experiments and standardized assays (e.g. scratch assay, tubule forming assay etc.), the cells in these essays are forced to migrate in rather artificial conditions. Not only is the movement confined to a flat surface, but also the stiffness of the migration substrate is often much higher than the extracellular matrices found in vivo. In addition, most of the chemical composition of the extracellular matrix is missing, effectively disabling molecular recognition mechanisms and invasive modes of migration; furthermore, there are generally no pores, which in turn disables most amoeboid modes of migration. Combine these shortcomings with the considerably smaller field of view of conventional microscopy ($\sim$1x1 mm$^2$, restricting the number of simultaneously observable cells to $10^2 - 10^3$), our method represents an improvement over all these aspects, while at the same being cheaper, easier and faster to handle.

Finally, we want to discuss some of our findings about the migration of neutrophil granulocytes. First off, our method is capable of replicating all of the known migratory peculiarities of neutrophils, namely (I) the right kind of response towards the chemokines IL-8 and fMLP, including the migratory direction, which follows the direction of the gradient; (II) the concentration dependency of these chemotactic responses, which passes through a maximum; (III) the right kind of dependency on pore size and stiffness of the collagen matrix, in particular the slowdown of cells with increasing collagen concentrations and the higher amount of turning events in these denser matrices. Notably, we found undirected cell speeds that were higher by a factor of $\sim$2–3 than in many other works about 3D and 2D neutrophil migration [26, 47, 50]. These authors consistently report undirected migration speeds of 6 μm to 8 μm/min, whereas we found values well above 20 μm/min. We have therefore double-checked our values for the undirected migration speeds independently using conventional phase contrast microscopy. These control experiments yielded similarly elevated migration speeds between $\sim$15 and 20 μm/min. Also, it has been reported by others that neutrophil migrations speeds can well exceed 20 μm/min (see e.g. [32, 51, 52]), which leads us to believe that the higher migration speeds found by us reflect the reality more accurately. To explain the differences found, it is important to first note that the works mentioned above give cell speeds averaged per cell over time periods of varying lengths, whereas we calculated ensemble averages for each time point of the experiment. Therefore, the momentary maximum speeds observed in those works might have been higher, though it remains unclear from the data presented. Though our measurements could be overestimating the real speeds due to the uncertainty in cell positioning (cf. Fig 2C), the effect of this noise is too small to account for the differences found. Also, manual tracking of single cells suggests that the effect is smaller for moving objects. Further possible reasons for the difference in cell speed to the above mentioned publications include: (a) the use of different cell types (in case of François et al.: neutrophil-like dHL-60 vs. freshly isolated neutrophils), (b) collagen of other origin and a different chemokine concentration, (b) lower frame acquisition rates: e.g. François et al. used a frame rate of 2 frames/min which, when applied to our time-series, would result in undersampling of the zig-zag movement found by us for the neutrophils. We suggest further experiments to resolve these questions because the real 3D migration speed of neutrophils is important for understanding the early stages of the innate immune reaction.

## Conclusion

In this paper, we present an important methodological advancement for investigating cell movement. In summary, we combined a lensless in-line holographic setup for 3D cell imaging with a very simple, yet flexible 3D migration assay, consisting of a collagen droplet submersed in cell culture medium in a small thin bottomed petri dish. Albeit very simple, this setup proved to be highly powerful for 3D-tracking thousands of unlabeled cells simultaneously. We employed open-source tracking algorithms to calculate cell tracks and to derive momentary and overall migration speeds, migration directedness and other important migration figures. We have used this setup to investigate neutrophils, the movements of which are particularly important in the response of the immune system towards pathogens and in inflammatory diseases. We could confirm previously reported migration patterns but with a considerable simplification of the experimental effort, while at the same time improving accuracy increasing the number of tracked cells by roughly one order of magnitude. In particular, we could demonstrate the applicability of our method to diagnose asthma from the reduced response of neutrophils to chemoattractants in asthma patients. This demonstrates the setup being applicable to novel diagnostic methods.

## Supporting information

**S1 Fig. Diffusion.** Diffusion gradient for rhodamin-labeled dextrane in collagen (MW 10 kDa) over a course of about 20 min, which was the duration of our tracking experiments. **(A)** Time series of fluorescence micrographs (5x, ZEISS LSM 510), displaying the border between the Rh-dextran solution (red upper part) and collagen (black lower part). **(B)** Intensity profiles of the above micrographs perpendicular to the diffusion interface, normalized to $I_0$. The diffusion gradient extends from 600 μm to 800 μm within 20 min.
(EPS)

**S2 Fig. Temperature distribution in the holographic setup.** Infrared image of our holographic setup right after a time-lapse recording of 60 min inside of an incubator at 37 degrees air temperature (IR camera FLIR P660). Four measuring points are displayed in the upper left box; the top reading of 37.5 degrees refers to the large crosshair and refers to the highest temperatures found within the petri dish. Therefore, the cells are not subjected to temperatures significantly higher than 37˚C.
(TIF)

**S3 Fig. x-y- and z-precision of the holographic setup.** Lateral (x-y-) precision has been determined using a calibration target (Carl Zeiss microscope stage micrometer). **(A)** displays the hologram (left) and magnitude reconstruction (right) of the stage micrometer, inset shows a zoom of the top end of the scale with ImageJ measuring tool. The length of the stage micrometer is 1 mm (±1 $\mu m$) and was determined to be 1042 $\mu m$ with our holographic setup, corresponding to a lateral precision of ±4.2%. **(B)** z-precision was measured using fluorescent micro-beads positioned below and on top of a glass cover slip submerged in water. We compared the z-positions determined with a conventional laser scanning microscope (Zeiss LSM 510) with those found in our holographic setup. The left image shows the reconstructed xz-view of the hologram (phase). The right image shows the same after 3D cross-correlation with a bead pattern where the brightest point is used to determine the z-position of a bead. **(C)** Comparison of measured z-positions of beads below and above the glass cover slip, measured with our holographic device (left) and a laser scanning microscope (right), respectively. Z-positions of the holographic setup have a slightly higher standard deviation than the those of the reference, but the overall accuracy of the setup is within ∼6% of the reference

microscope (LSM: 120.9$\mu m$ average z-distance between upper and lower beads / holographic setup: 127.9$\mu m$).
(EPS)

**S4 Fig. Evaluation of the tracking algorithm.** A ground truth was created manually (see S1 Dataset). We determined (1) the number of false positive cells; (2) the number of false negative cells and (3) the average distance of the tracked cells to the ground truth. The number of false positive cells was determined as the number of cells found by the tracking algorithm for which no neighbour could be found in the ground truth within a 11.2μm distance. This search radius has been chosen because it is also the search radius of our tracking algorithm. Conversely, every cell in the ground truth, for which no tracked cell was found within 11.2μm, was counted as a false negative. The numbers are: Total cells in ground truth: 8105; total cells found by algorithm: 8046; Total false positives: 775 (9.6%); Total false negatives: 484 (6.0%). Additionally, the average distance between true and computed tracks was determined by comparing the distance distributions of nearest neighbors within the ground truth to that between ground truth and tracked cell positions (i.e. the average distance of the tracked cells to the ground truth). The **(left)** histogram displays the distance distribution of cells within the ground truth (average distance 26.6 μm), while the **(right)** histogram displays the distance distribution between cells in the ground truth and the tracked cell positions (average distance to be 4.2 μm). Therefore, a tracked cell will be generally much closer to the true cell position than to a (false) neighboring cell. These measurements highlight the importance of using low to medium cell densities ($10^3$-$10^6$ cells/ml)) for achieving good tracking results.
(EPS)

**S5 Fig. Comparison of chemotatic velocities of asthmatic vs non-asthmatic individuals.** We compared the migration velocity towards the gradient of IL-8- and fMLP-stimulated neutrophils with their respective control groups (n = 3 in every of the four groups). To do so, we averaged over 75 values of the outward migration velocities starting after 2.5 min (i.e. 5 minutes of tracking, leaving out the onset of migration, cf. Fig 7A and 7B). The result of these comparisons is displayed for IL-8 **(left)** and fMLP **(right)**. It can be seen that IL-8 might be better suited than fMLP for telling apart asthma from non-asthma-conditions (p-values of 0.03* and 0.09 for IL-8 and fMLP, respectively).
(PDF)

**S1 Movie. Phase shift minimum projection of neutrophil migration under IL-8 stimulation.** This video shows a time lapse of a typical 20 min migration experiment (sped up 120 x) where 100 ng/ml IL-8 had been added to the surrounding medium to stimulate neutrophil migration. Shown is a crop of the phase shift minimum projection (cells are dark spots) with roughly a quarter of the collagen drop visible.
(AVI)

**S1 Dataset. Ground truth for evaluation of the tracking algorithm.** Ground truth for evaluation of the tracking algorithm. A subset of 50 frames (about 3.5 min) and size 581x562 pixel (650x630 um) was manually annotated in the phase shift minimum projections of a time-lapse recording of neutrophils (stimulation 100 ng/ml IL-8). For results of benchmarking our algorithm see S4 Fig.
(ZIP)

## Author Contributions

**Conceptualization:** Falk Nette, Ana Cristina Guerra de Souza, Daniel Hans Rapoport.

**Data curation:** Falk Nette.

**Formal analysis:** Ana Cristina Guerra de Souza.

**Funding acquisition:** Daniel Hans Rapoport.

**Investigation:** Falk Nette, Ana Cristina Guerra de Souza, Tamás Laskay, Mareike Ohms, Daniel Dömer, Daniel Drömann, Daniel Hans Rapoport.

**Methodology:** Falk Nette, Ana Cristina Guerra de Souza, Daniel Hans Rapoport.

**Project administration:** Daniel Hans Rapoport.

**Resources:** Tamás Laskay, Mareike Ohms, Daniel Dömer, Daniel Drömann, Daniel Hans Rapoport.

**Software:** Falk Nette.

**Supervision:** Daniel Hans Rapoport.

**Visualization:** Falk Nette.

**Writing – original draft:** Falk Nette, Ana Cristina Guerra de Souza, Daniel Hans Rapoport.

**Writing – review & editing:** Falk Nette, Ana Cristina Guerra de Souza, Tamás Laskay, Mareike Ohms, Daniel Dömer, Daniel Drömann, Daniel Hans Rapoport.

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
