## [Decision Letter · Decision Letter 0]

25 Apr 2022

PONE-D-22-08894Method for simultaneous tracking of thousands of unlabeled cells within a transparent 3D matrixPLOS ONE

Dear Dr. Rapoport,

Thank you for submitting your manuscript to PLOS ONE. After careful consideration, we feel that it has merit but does not fully meet PLOS ONE’s publication criteria as it currently stands. Therefore, we invite you to submit a revised version of the manuscript that addresses the points raised during the review process.

We look forward to receiving your revised manuscript.

Kind regards,

Kun Chen, Ph.D

Academic Editor

PLOS ONE

Journal Requirements:

[This work was partly funded by the European Regional Development Fund (ERDF).]

 [Work carried out by FN and ACGS was funded by the European Regional Development Fund (https://ec.europa.eu/regional_policy/en/funding/erdf/) via the FIT-program of Schleswig-Holstein, Germany („KillAsthma“, Grant-Number 123 17 024). The funders had no role in study design, data collection and analysis, decision to publish, or preparation of the manuscript.]

Reviewers' comments:

Reviewer's Responses to Questions

**Comments to the Author**

1. Is the manuscript technically sound, and do the data support the conclusions?

Reviewer #1: No

Reviewer #2: Yes

Reviewer #3: Yes

2. Has the statistical analysis been performed appropriately and rigorously? 

Reviewer #1: No

Reviewer #2: N/A

Reviewer #3: Yes

3. Have the authors made all data underlying the findings in their manuscript fully available?

Reviewer #1: Yes

Reviewer #2: Yes

Reviewer #3: No

4. Is the manuscript presented in an intelligible fashion and written in standard English?

Reviewer #1: Yes

Reviewer #2: Yes

Reviewer #3: Yes

5. Review Comments to the Author

Reviewer #1: This publication describes a new methodology to track a large number of cells in a 3D hydrogel. In particular, the tracking of immune cells with a large statistic (N>3000 cells). The methodology is based on lens-free microscopy as first developed in Ozcan group and/or Allier group. The authors shows that lens-free microscopy allows to follow in 3D a very large number of cells with a high sampling rate (one acquisition every 4 seconds) and a micrometric precision (1.5µm in X-Y and 30µm in Z). Several case studies describes the experiments performed with neutrophil cells from different donors under different matrix and/or drug conditions.

Major revision

1/ The evaluation of the measurements of cell detection, cell position and cell tracking is not well described, it is not supported by proper measurements and proper calculations. I strongly encourage the author to improve these assessments to better characterize the methodology. Cell detection should be evaluated in terms of precision and recall calculated from comparison measurements made with a reference measurement, i.e., another reference microscope, such as fluorescence microscopy. Similarly, cell position should be assessed with precision measurements obtained by comparing lens-free microscopy to another reference microscopy. Here, the authors simply state "we estimate that the accuracy that can be obtained from these reconstructions is 30 µm in Z position and 1.5 µm in X and Y position." This is not a robust assessment. As an example, please follow the methodology discussed in the supplementary information of 'Berdeu et al.' Finally the cell tracking efficiency should be assessed with a given tracking index calculated with a series of annotated ground truth cell tracking. The methodology is well described in 'Ulman, V., Maška, M., Magnusson, K. E., Ronneberger, O., Haubold, C., Harder, N., ... & Ortiz-de-Solorzano, C. (2017). An objective comparison of cell-tracking algorithms. Nature methods, 14(12), 1141-1152.'. I disagree with the authors when they write ‘Tracking reliability can be assessed quantitatively by analyzing the distribution of 339 track lengths.’.

In sum, from my point of view further assessments are needed, they are very important to characterize the overall methodology discussed in this paper.

2/ I have a concern dealing with the temperature control of the cell culture. If the CMOS sensor is very close to the cell and if it is too hot or not stable in temperature, it can influence the cell culture condition and consequently the cell measurements. This point is discussed in the conclusion, but there is no mention of this point in the Material and methods part nor in the Results part. I would suggest the author to conduct a thermal assay and present proper results to better ensure that CMOS heating is not influencing the culture conditions over time.

3/ The author point that they demonstrate the applicability of their method to diagnose asthma. I disagree to assess a diagnostics tools a stronger study with a cohort of patients and a sound experimental plan is needed. This is beyond the scope of the present paper. Here the author discuss the comparison between a sample from one ‘asthmatic’ donor and another sample from a ‘non-asthmatic’ : this is a single comparison experiment (Fig 7). I would suggest the author to perform repetability and reproducibility experiments to ensure that the difference found can serve as a proper signal for a diagnostic tool.

Minor comments

- In the abstract, change ‘the most powerful methods’ to ‘a powerful method’.

- In the abstract, change ‘spatial resolution’ to ‘position accuracy’

- In the introduction, the author mention that the study of immune responses require precise knowledge of thousand cells. They could cite and discuss this interesting study to highlight the requisites to study immune response ‘Crainiciuc, G., Palomino-Segura, M., Molina-Moreno, M., Sicilia, J., Aragones, D. G., Li, J. L. Y., ... & Hidalgo, A. (2022). Behavioural immune landscapes of inflammation. Nature, 1-7.’, a sound study

- Line 22 ‘when compared previous methods,’, Which methods ?

- Line 32-34 : ‘The portion of the light which passes through the cells becomes phase-shifted and diffracted with respect to the light which did not go through the cells (reference beam). Superposition of these different light paths yields diffraction patterns which can be recorded in the near-field of the sample’. This is uncorrect, there is no reference beam, light paths do not superpose, please rewrite the sentences.

- Line 38 cite a reference dealing with holographic reconstruction

- Line 177 cite the reference of the CMOS sensor

- Line 178 what is ‘a true resolution’ ?

- Line 256 how cell segmentation is perfomed ? what cell metrics are computed ?

- Line 503 what is inexpensive ?

Reviewer #2: This manuscript demonstrated a very interesting application of a lensless in-line holography in label-free tracking thousands of unlabeled cells within 3D collagen matrix. The principle of technique used in this manuscript is basically the smart integration between two well-established approaches. The lensless DHM with LED illumination provides a relative good lateral resolving power and easy implementation, enabling 3D particles/cells tracking from a single digitally recorded hologram. The practice of minimum z-projection improves the tracking accuracy. Around 3000 human neutrophil granulocytes can be tracked simultaneously with a single-shot within a large FOV with high spatiotemporal resolution, even though axial resolution inherently suffers from limitations same to other in-line holography. In general, this manuscript illustrated a very detailed and inspiring example of lensless in-line holography. The logic of experimental design and validation are impressive and solid. I hold a positive opinion for publishing this manuscript on PLOS ONE journal, with some minor comments to the manuscript.

1. I didn’t get the motivations for measuring neutrophil migration speed under different control experiments. I am wondering if authors could explain more about the motivation and what kind of hints that could be biologically meaningful as a simple take-away message or future directions.

2. Authors mention using ‘minimum phase projection’ along z for extracting axial coordinates. Do you mean ‘minimum intensity projection’? Most of in-line holography can do minimum intensity projection for z axial tracking, but phase projection is not very common.

3, Three axes coordinates were measured but only lateral axes were used in studies. During the experiment measurements, z information was also recorded. However, when calculating average undirected speed and average velocity of all cells outward collagen droplet, only lateral coordinates were used. In authors self-explanation, it might be attributed to 20-30um axial resolution is not good enough for such a tracking purpose. But assuming migration along three axes are independent from each other, as shown in Fig. 3B, the migration distance can be as large as tens of um, which is in the resolvable orders of system’s axial resolution. Since the sample is a 3D matrix, it might be very interesting to include the information from z direction, as hinted by a similar experiments, however, targeting at sperm migration where z directional migration shows unique sperm swimming patterns (Su, Ting-Wei, PNAS, 109 (40), 2012).

4, Nonlinear response in Fig. 5A. The average unidirectional speed is quite different in different collagen concentrations. Temporal nonlinear response is expected. However, I am wondering if authors might have some additional explanations that why in both figures in Fig. 5A, the flatten or maximum of s or v is not ordered or proportional to collagen concentration, but instead, is very nonlinear. How many repetitions or collagen samples that the experiments has done? Also, it might be attributed to the conditions of sample making, like temperature, polymerization time. Similar unusual observation can also be seen in Fig. 5B, where 1mg/mL is way too higher than any other concentrations.

5, During discussion part, in Line 513-520, I don’t think that the large FOV and high frame rate are the unique strength or improvement of current system, compared with other similar in-line holography. In my understanding, the difference just merely results from the hardware configuration (like just change to a large camera), without fundamental breakthrough of the technique. If author believes so, it might be more convincing to discuss more about the improvements of current apparatus compared to previous ones.

6, I am also curious that why the collagen has to be droplet. Can that also be cells cultured in petri-dish or embedded in collage 3D matrix with petri-dish? I believe the latter could broader the interest of readers.

7, I would recommend authors to do more statistical analysis of the data to show the significant difference between different migration patterns. This will make the scientific story more convincing.

8, some typo errors. Like, line 516, 20,000 rather than 20.000. Fig. 5B and 5C should switch their labels in the main text.

Reviewer #3: This paper presents a handy methodology to study 3d cell migration in vitro. Remarkable size of the field of view, num. of cells, spatial/temporal resolution are achieved. Validity is demonstrated by studying chemotaxis and behavioral changes of neutrophils with IL-8, fMLP, upon treatment with LPS or isolation from patients with asthma. I consider these appropriate and relevant applications. Moreover, neutrophilsare the ideal cell type for validation due to their plasticity, possibly non-convex shapes, and high migration speed.

My main concerns:

1. A common problem of collagen is the formation of bubbles during solidification (see Fig5 PMC7757667). Is it possible to recognize bubbles in the min projections and eventually decide to discard the sample? Or is it suggested to check also the diffraction images or inspect sample with another method before imaging?

2. Not clear if the expected cell diameter should be known for digital reconstruction. Please.also explain if the system can be used to image together cell types of different size (i.e. neutrophils and dendritic cells).

3. Fig1, a histogram with the number of cells detected at different z depths vs. the expected number could clarify how cell detection is robust to positioning along z.

4. Special care was put to create a time series of minimum projections and depth maps for neutrophils. This was achieved by relying on Gouy phase anomaly, minimum projection, and an averaging scheme that accounts for non-globular cell shapes. In L233, Fig2A it is not clear to me which is the connected foreground component. Assuming the black dots.

5. Min proj. and depth map saved spage. However, please clarify if cells lying on top of the other along z be resolved as separate objects.

6. Although the authors assessed the formation and stability of gradients within the droplet, it would be interesting to know in Fig3A if certain regions have higher motility (i.e. at the droplet boundaries due to fiber orientation?) A motility heatmap computed using optical flow (like Fig6C, PMC6881817), or maximum speed at different locations may show this.

7. I recommend depositing raw data in an open bioimage repository, if possible. Diffraction images can be useful to others developing computational methods for reconstruction, while reconstructed videos can be helpful to develop analysis tools.

8. L78 missing reference

6. PLOS authors have the option to publish the peer review history of their article (what does this mean?). If published, this will include your full peer review and any attached files.

Reviewer #1: No

Reviewer #2: **Yes: **Yuechuan Lin

Reviewer #3: **Yes: **Diego Ulisse Pizzagalli

---

## [Author Response · Author response to Decision Letter 0]

7 Jun 2022

We have uploaded our response to reviewers as a Word file (Response_to_Reviewers.docx).

---

## [Editor Report · Decision Letter 1]

10 Jun 2022

Method for simultaneous tracking of thousands of unlabeled cells within a transparent 3D matrix

PONE-D-22-08894R1

Dear Dr. Rapoport,

We’re pleased to inform you that your manuscript has been judged scientifically suitable for publication and will be formally accepted for publication once it meets all outstanding technical requirements.

Kind regards,

Kun Chen, Ph.D

Academic Editor

PLOS ONE
---

## [Editor Report · Acceptance letter]

16 Jun 2022

PONE-D-22-08894R1 

Method for simultaneous tracking of thousands of unlabeled cells within a transparent 3D matrix 

Dear Dr. Rapoport:

I'm pleased to inform you that your manuscript has been deemed suitable for publication in PLOS ONE. Congratulations! Your manuscript is now with our production department. 

Kind regards, 

on behalf of

Dr. Kun Chen 

Academic Editor

PLOS ONE